

# Identifying topology of distribution substation in power Internet of Things using dynamic voltage load fluctuation flow analysis

Yongjin Xu[1], Jifan Lv[1], Jiaying Wang[1], Fangbin Ye[1], Shen Ye[1] and Jianfeng Ji[2]

[1] State Grid Zhejiang Marketing Service Center, Hangzhou, China
[2] Beijing Zhixiang Technology Co., Ltd, Beijing, China

## ABSTRACT

At present, the reconfiguration, maintenance, and review of power lines play a pivotal role in maintaining the stability of electrical grid operations and ensuring the accuracy of electrical energy measurements. These essential tasks not only guarantee the uninterrupted functioning of the power system, thereby improving the reliability of the electricity supply but also facilitate precise electricity consumption measurement. In view of these considerations, this article endeavors to address the challenges posed by power line restructuring, maintenance, and inspections on the stability of power grid operations and the accuracy of energy metering. To accomplish this goal, this article introduces an enhanced methodology based on the hidden Markov model (HMM) for identifying the topology of distribution substations. This approach involves a thorough analysis of the characteristic topology structures found in low-voltage distribution network (LVDN) substations. A topology identification model is also developed for LVDN substations by leveraging time series data of electricity consumption measurements and adhering to the principles of energy conservation. The HMM is employed to streamline the dimensionality of the electricity consumption data matrix, thereby transforming the topology identification challenge of LVDN substations into a solvable convex optimization problem. Experimental results substantiate the effectiveness of the proposed model, with convergence to minimal error achieved after a mere 50 iterations for long time series data. Notably, the method attains an impressive discriminative accuracy of 0.9 while incurring only a modest increase in computational time, requiring a mere 35.1 milliseconds. By comparison, the full-day data analysis method exhibits the shortest computational time at 16.1 milliseconds but falls short of achieving the desired accuracy level of 0.9. Meanwhile, the sliding time window analysis method achieves the highest accuracy of 0.95 but at the cost of a 50-fold increase in computational time compared to the proposed method. Furthermore, the algorithm reported here excels in terms of energy efficiency (0.89) and load balancing (0.85). In summary, the proposed methodology outperforms alternative approaches across a spectrum of performance metrics. This article delivers valuable insights to the industry by fortifying the stability of power grid operations and elevating the precision of energy metering. The proposed approach serves as an effective solution to the challenges entailed by power line restructuring, maintenance, and inspections.

Corresponding author
Jiaying Wang,
wangjiayingee@zju.edu.cn

# INTRODUCTION

At present, the restructuring, maintenance, and inspection of power lines hold paramount significance in ensuring the stability of grid operations and the accuracy of electrical energy measurement. These critical operations guarantee the power system's seamless functioning, enhance the power supply's reliability, and contribute to precise electrical energy consumption measurement. Furthermore, the importance of power grid topology identification cannot be overstated, as it directly affects grid stability and the accuracy of electrical energy measurement. Traditional power lines are exposed to various operational factors, such as capacitance and inductance, which can lead to power signal transmission instability, thereby impacting grid reliability. Additionally, power line structures' topology plays a pivotal role in grid management and maintenance, as it determines key functions like power flow, fault detection, and isolation. Nevertheless, current power line topology identification (PLTI) encounters challenges, including the influence of grid components on conventional methods based on power line carrier communication, leading to increased costs and noise interference. Furthermore, issues related to topology identification in low-voltage distribution network (LVDN) substations arise within the context of smart distribution grid development. Topology identification plays a pivotal role in the realm of energy management systems and distribution management systems by enabling the mathematical modeling of a power network's physical structure for subsequent analysis and computation. It furnishes crucial network structure data indispensable for various functions within the power system, including power flow calculation, state estimation, dynamic security analysis, fault analysis, and reactive power optimization, among others (*Xie et al., 2022*; *Srinivas & Wu, 2022*). The advent of the Power Internet of Things (PIoT) has ushered in a transformative era within the power system, where diverse electrical devices and personnel are interconnected through information sensing devices. This convergence of network information technology, electronic technology, and artificial intelligence has given rise to a vast, intelligent network that transcends multiple domains. In the context of LVDNs, several challenges emerge owing to their intricate structure and the limited technical resources available for detecting and verifying network topology anomalies. Present-day maintenance practices heavily rely on manual methods, leading to delays in updates and compromising data quality. Consequently, a significant issue arises regarding the misalignment between the physical topological model and the information model and data housed within power information systems. Therefore, a central challenge in the realm of PIoT lies in the development of technical mechanisms for validating the topological structure of LVDNs during operation, thereby effectively addressing the challenges associated with information and data sharing (*Blackmore et al., 2021*; *Klein, Oppelt & Kuenzer, 2021*).

Scholars specializing in PLTI have conducted extensive research, providing valuable insights into the operation and management of power systems (*Shah & Zhao, 2022*). This

section highlights recent studies pertinent to this article, delving into their advancements, merits, and constraints. *Shahdadian et al. (2022)* introduced a deep learning-based approach for PLTI. Their method leverages convolutional neural networks (CNNs) and long short-term memory networks to extract topological features and temporal correlations. While this approach demonstrates commendable accuracy and robustness, it entails computational complexity, particularly in large-scale power systems. *Chakraborty, Jain & Seo (2022)* explored a voltage topology identification method founded on support vector machines (SVM). By constructing feature vectors from voltage measurement data and deploying SVM models, they achieved noteworthy classification performance and computational efficiency. However, the accuracy of this method may be constrained when dealing with intricate power system structures. In addressing optimization challenges within PLTI, *Chang et al. (2023)* proposed a genetic algorithm-based methodology. Their approach employs genetic algorithms to search for the optimal topology structure, corroborated by current injection measurement data. This strategy attains high accuracy and scalability in topology identification, although computational complexity remains a concern in large-scale power systems. Conversely, *Flynn et al. (2023)* presented a graph theory-based PLTI method. Their approach employs graph theory algorithms to establish relationships among power system nodes and edges, showcasing commendable scalability and computational efficiency. Nevertheless, accuracy challenges may arise in power systems characterized by complex loops or multiple paths. In summary, these recent studies represent significant advancements in the field of power network and voltage topology identification. Each approach possesses distinct strengths and limitations, offering diverse solutions and perspectives for topology identification in power systems. However, it is essential to acknowledge potential accuracy limitations, particularly in the context of complex system structures. This article endeavors to introduce a novel method that combines the Internet of Things (IoT) technology, topology analysis algorithms, and dynamic voltage load fluctuation analysis to enhance the accuracy and efficiency of conventional methods. This approach aims to provide robust support for the operation and stability of distribution substations.

In summary, traditional substation topology identification systems primarily rely on power line carrier communication, employing techniques such as spread spectrum, frequency modulation, intermediate frequency, and audio carrier to transmit signals. However, in such scenarios, the presence of capacitance and inductance within the power grid significantly impacts the stability of carrier signal transmission. This necessitates the deployment of relay equipment, wave traps, and filters, consequently increasing costs and rendering the power grid more susceptible to noise interference. Importantly, the inability to transmit communication signals over the power lines compromises the accuracy of substation topology identification. Distribution substations constitute a pivotal component of intelligent distribution networks, ushering in new challenges related to power quality, power supply reliability, and economic operations. In comparison to the transmission network, the intelligent construction level of the distribution network lags behind. Similarly, the LVDN trails behind the medium voltage distribution network in terms of intelligent infrastructure. Consequently, LVDN emerges as a vulnerable aspect in smart

grid development. Management approaches in this domain tend to be relatively simplistic, lacking intelligent mechanisms for holistic, effective, and precise management of LVDN. Consequently, there exists an urgent need for technological advancements to monitor and support the comprehensive operation of LVDN substations. This article introduces an innovative approach to topology identification in LVDN substations, rooted in voltage correlation analysis employing dynamic time series segmentation (TSS). The proposed method leverages the extreme point determination (EPD) technique for time series data processing (*García et al., 2023*). Building upon this foundation, the voltage correlation principle is applied to assess the voltage correlation across different sequences, including extreme points, facilitating the identification of the substation's topology structure. The strength of this article lies in its augmentation of the time series algorithm through the incorporation of a TSS algorithm based on the constrained hidden Markov model (HMM). The TSS algorithm evolves from the optimization of the time series algorithm. Specifically, it integrates constrained HMM techniques to achieve more precise segmentation of time series data, leading to enhanced topology identification accuracy. These enhancements and optimizations bolster the adaptability and performance of the TSS algorithm in practical applications.

This article contributes significantly by proposing a method for topology identification in LVDN substations based on voltage correlation analysis. It achieves more precise TSS and topology identification through the introduction of the TSS algorithm, which is based on a Constrained HMM. The importance of this research lies in its capacity to address the challenges of topology identification in LVDN substations within the context of smart distribution grid development. It enhances the management and maintenance of the power grid and improves the accuracy of electrical energy measurement, thereby offering practical technical support for the stable operation of smart grids. The power grid's adaptability to changes and faults can be improved by optimizing topology identification methods, subsequently enhancing the reliability of the power supply. This research holds significant potential implications for the sustainable development of smart distribution grids and electrical energy management.

## LITERATURE REVIEW

The current state of research on PLTI, both nationally and internationally, underscores the continuous development and significance of this field. Internationally, cutting-edge technologies include the application of methodologies such as power line carrier communication, smart sensors, data analysis, machine learning, and artificial intelligence. These methodologies have the potential to improve the stability of power grids and the accuracy of electrical energy measurement. *Farajollahi, Shahsavari & Mohsenian-Rad (2019)* provided insights into the latest international advancements in power grid topology identification. They explored various traditional and state-of-the-art methods for topology identification, such as power line carrier communication and smart sensors, highlighting their applications in enhancing power grid stability and precision in electrical energy measurement (*Farajollahi, Shahsavari & Mohsenian-Rad, 2019*). *Deka, Talukdar & Chertkov (2020)* summarized the state-of-the-art technologies employed for topology

identification in international smart grid contexts. They introduced methodologies based on data analysis, machine learning, and artificial intelligence, elucidating how these approaches contribute to the stability of power grids and the accuracy of electrical energy measurement (*Deka, Talukdar & Chertkov, 2020*). *Ramakrishna & Scaglione (2021)* discussed how international researchers leverage advanced technologies to gain a deeper understanding of power system structures and behaviors, enabling improved monitoring and management of power grids. Moreover, the article emphasized the potential of these technologies to promote grid intelligence and create new opportunities for sustainable power supply (*Ramakrishna & Scaglione, 2021*). *Hosseini, Khodaei & Paaso (2020)* proposed topology identification techniques tailored for distributed power grids. This approach incorporates methodologies such as time series analysis and constrained hidden Markov models to enhance power grid management and maintenance (*Hosseini, Khodaei & Paaso, 2020*).

In China, researchers are actively contributing to the enhancement of power system stability and supply reliability through advancements in current phase detection and data mining technologies. *Zhang, Guo & Zheng (2022)* introduced a method for identifying the topology of LVDN substations based on voltage correlation analysis. This approach specifically targets topology identification challenges within LVDNs by employing EPD of time series voltage data. Subsequently, voltage correlation analysis is applied to different time series data to discern the substation's topology structure (*Zhang, Guo & Zheng, 2022*). *Li, Wang & Wang (2021)* devised a PLTI method grounded in SVMs. The researchers employed the SVM algorithm to analyze voltage and current data from power lines and trained the model to recognize topology structures. This method demonstrates high accuracy and robustness, rendering it suitable for intricate power grids (*Li, Wang & Wang, 2021*). *Zhang, Shi & Zhang (2022)* introduced a deep learning-based algorithm for power grid topology identification. Their approach leverages CNNs and recurrent neural networks to process power grid data, enabling automatic detection and identification of topology structures (*Zhang, Shi & Zhang, 2022*). *Yin, Zhu & Hu (2019)* proposed a topology structure optimization method based on genetic algorithms to enhance the precision of PLTI. This algorithm harnesses the search capabilities of genetic algorithms to tackle complex challenges in topology identification (*Yin, Zhu & Hu, 2019*). *Shen, Cui & Wang (2020)* presented a fuzzy logic-based approach for power system topology identification. They employed fuzzy logic to address uncertainty and ambiguity in power systems' information, facilitating the identification of power grid topology structures (*Shen, Cui & Wang, 2020*).

In conclusion, prior research on PLTI has made significant strides; however, it still grapples with certain limitations, including inadequate consideration of power system complexity, precision shortcomings, and insufficient algorithmic robustness. Hence, the distinctiveness of this study is twofold: First, it introduces a topology identification method grounded in voltage correlation analysis, enabling a comprehensive understanding of complex relationships and variations within the power system. Second, it integrates the constrained hidden Markov model for a more precise analysis of voltage data, thereby enhancing the accuracy of topology identification and bolstering the resilience of time

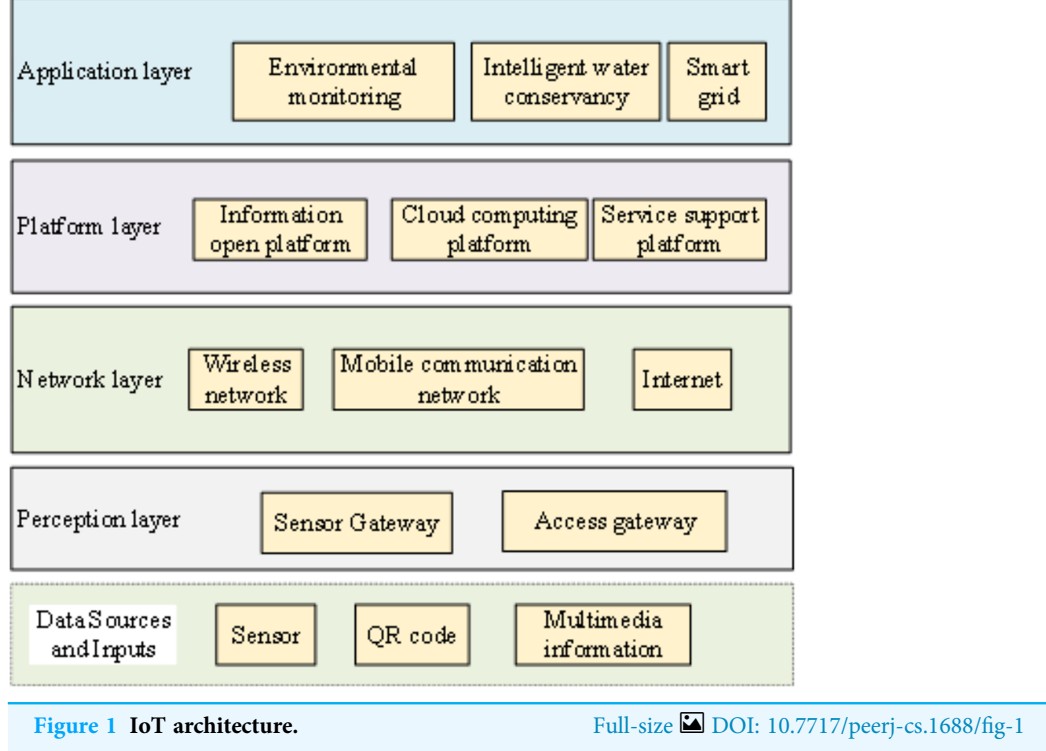

**Figure 1 IoT architecture.** 

series analysis. This approach is anticipated to yield superior adaptability and performance in real-world applications, ultimately elevating power system stability, enhancing energy metering accuracy, and furnishing valuable technical support for power system monitoring and maintenance.

# SUBSTATION TOPOLOGY IDENTIFICATION AND PIOT TECHNOLOGY

## IoT and PIoT

The IoT constitutes a network facilitating information exchange among objects and individuals over the internet. It has emerged as a prominent focal point in the next generation of the information revolution, garnering worldwide attention (*Chan et al., 2019*; *Park, Lee & Choi, 2019*). Its applications span a wide array of domains, including smart cities, intelligent transportation, and telemedicine. Within this spectrum, IoT's integration in the realm of electrical power, as a significant branch of IoT, assumes a pivotal role within power systems (*Wang et al., 2021*). The IoT architecture is shown in Fig. 1.

PIoT interconnects diverse electrical devices and personnel within the power system using information sensing devices. It amalgamates network information technology, electronics, and artificial intelligence to augment the reliability, efficiency, and intelligence of the power grid (*Sharma et al., 2021*; *Aguilera, Ortiz & Ortiz, 2021*). For example, PIoT facilitates applications such as smart power services, energy efficiency management, and intelligent home control, offering users a more user-friendly electricity experience. Moreover, it enhances load balancing and increases energy resource utilization efficiency within the power system (*Rodrigues et al., 2022*). These practical illustrations underscore

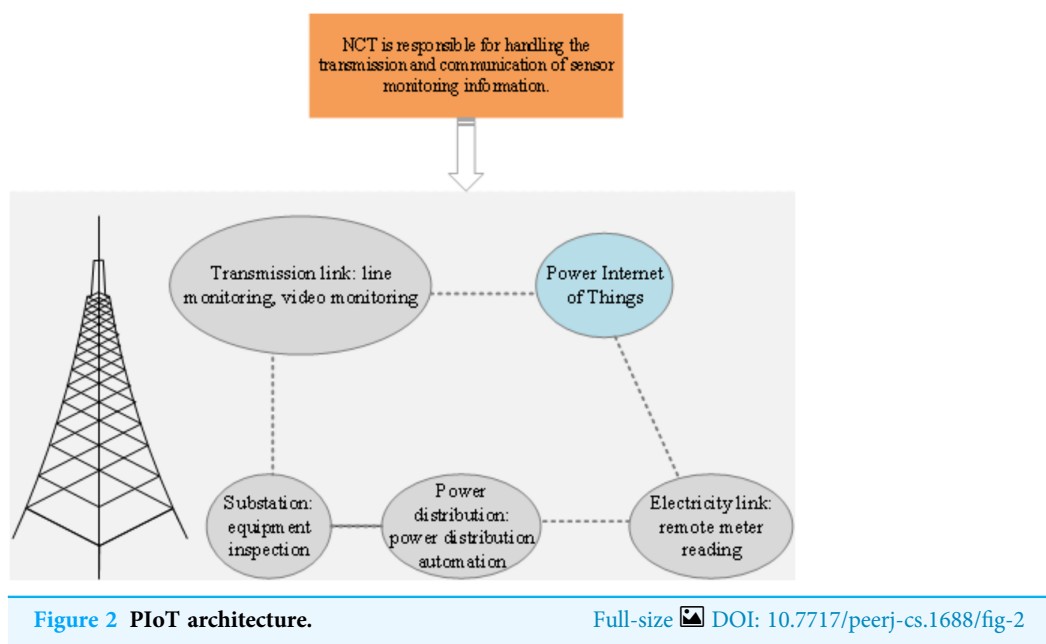

**Figure 2  PIoT architecture.**               

the significance and potential advantages of the PIoT in power systems. A comprehensive PIoT encompasses communication transmission technology, information processing, big data technology, cloud computing, and various data management technologies, thereby evolving into an intricate intelligent network ecosystem (*Pang, 2020*). The overarching framework of the PIoT is depicted in Fig. 2.

Figure 2 illustrates the architectural framework of the PIoT, comprising five interconnected components: transmission links (including line monitoring and video surveillance), PIoT, power lines (for remote meter reading), distribution systems (enabling distribution automation), and substations (facilitating equipment maintenance). At the core of the PIoT architecture lies network communication technology, which serves as the foundational infrastructure for transmitting and communicating sensor monitoring data. Network communication technology establishes connections between sensor devices, whether through wireless or wired networks, facilitating the transfer of sensed information to the PIoT. The PIoT assumes the pivotal role of a central hub for data aggregation and processing. It meticulously gathers and manages data originating from various sensors and devices, thereby offering real-time monitoring and vital support for power system operation and management. Leveraging advanced data processing and analysis techniques, the PIoT processes sensor monitoring data, enabling real-time monitoring, analysis, and diagnostics. This capability empowers the extraction of valuable insights and information, subsequently underpinning decision-making processes and operational management within the power system.

The electronic information system of the PIoT encompasses a variety of technologies drawn from diverse domains. These technologies frequently manifest in unique technical configurations and application specifications across various industries. Figure 3 elucidates the architectural model of PIoT technology through four distinct modules (*Mabrouki et al., 2021*; *Sharma, Sharma & Ahlawat, 2021*).

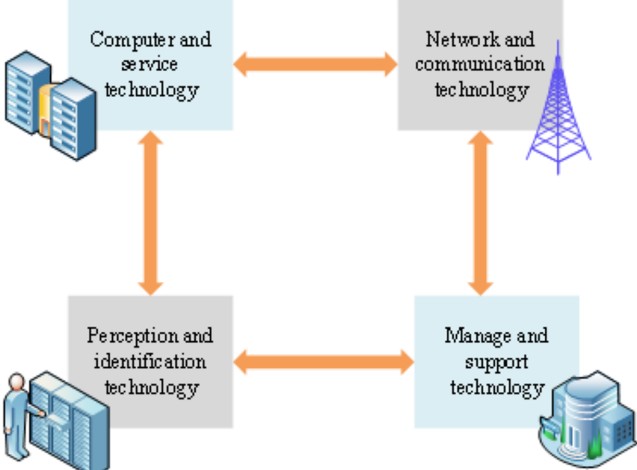

**Figure 3 Architecture model of PIoT technology.**

Figure 3 comprises four pivotal modules: perception and identification technology, network and communication technology (*Yang & Wang, 2021*; *Martins et al., 2021*), computing and service technology, and management and support technology (*Chen et al., 2021*; *Paudel & Neupane, 2021*). These modules assume critical roles in the system's functionality and operational dynamics.

## Voltage correlation based on time series analysis

Voltage correlation is widely utilized in time series analysis as a crucial quantitative measure for determining power system topologies. The fundamental principle of voltage correlation can be understood as follows: Voltage drop occurs when current flows through the system impedance, resulting in voltage waveforms that exhibit similar sequences at various points along the system impedance. The degree of electrical coupling between two measurement points can be assessed by analyzing the similarity of voltage variation waveforms. Common quantitative measures for this purpose include the Euclidean distance, dynamic time warping, and Pearson correlation coefficient. Notably, while Euclidean distance and dynamic time warping lack a normalized basis for judgment, necessitating comparative data analysis for assessment, the Pearson correlation coefficient offers a normalized measure.

The Pearson correlation coefficient (PCC), denoted as $R(X, Y)$, represents a statistical measure ($\in [-1, 1]$) quantifying the extent of linear correlation between two sequences, $X$ and $Y$. The magnitude of PCC signifies the degree of linear correlation, with a larger absolute value indicating a stronger correlation. A negative PCC signifies a negative correlation. The precise computation of PCC is as Eq. (1).

$$R(X, Y) = \frac{\text{cov}(X, Y)}{\sigma X \sigma Y} \tag{1}$$

In Eq. (1), $cov(X, Y)$ represents the covariance between one sequence, while $\sigma X$ and $\sigma Y$ denote the standard deviation of sequences $X$ and $Y$, respectively. This article utilizes the PCC to assess the relationship between users and stations. Initially, a threshold $\delta$ is established. If the calculated correlation, $R(X, Y)$, surpasses $\delta$, it signifies a robust correlation between the two sequences, even when there is a substantial disparity. In such instances, the topological connection between the user and the Substation is deemed accurate. Conversely, if $RR(X, Y) < \delta$, it indicates a weak correlation between the two sequences, implying an error in the topological connection between the user and the Substation, necessitating on-site verification.

The TSS algorithm serves two distinct applications: 1. The TSS algorithm can identify instances when model parameters or the model itself undergo changes. 2. TSS can generate a higher-level time series representation, applicable for indexing, clustering, classification, inconsistency discovery, and anomaly detection within time series data. These two application scenarios impose specific requirements on the TSS algorithm. In Scenario 1, the TSS algorithm must possess the capability to identify the time instances when changes occur in the system model. In Scenario 2, the TSS algorithm divides time series data into non-overlapping subsequences. In the context of this study, the voltage time series of users is segmented to enhance the precision and speed of topology verification. Subsequently, these subsequences are extracted to provide a more robust representation of users' power consumption characteristics. Building on this foundation, the article delves into the correlation between the substation and the voltage time series of users.

## HMM

The HMM stands as a widely employed probabilistic model for the analysis of sequential data. In this study, the HMM model is harnessed to address the issue of PLTI. The HMM model effectively captures and infers the topological states within power systems by encapsulating the statistical relationship between hidden states and observable sequences. This approach contributes to an improved understanding and analysis of the topological characteristics and variations among power network nodes. To align with the research objectives, a constraint is introduced in the form of the minimum continuous state length associated with change points. This constraint serves to regulate state transitions within the HMM model, ensuring its alignment with actual power systems. Subsequent sections of this article provide a comprehensive elucidation of the construction and optimization process of the constrained HMM, along with its practical application in the realm of PLTI.

HMMs can be categorized into two types: first-order HMMs and higher-order HMMs. HMM represents a dual random process where the transition from one state to another follows a random pattern, constituting an implicit finite-state Markov chain. Concurrently, the observations within a state are also subject to randomness, establishing a statistical correlation between the state and the observations. Figure 4 illustrates the underlying principle of HMM. The mathematical formulation of HMM is articulated through Eqs. (2) and (3).
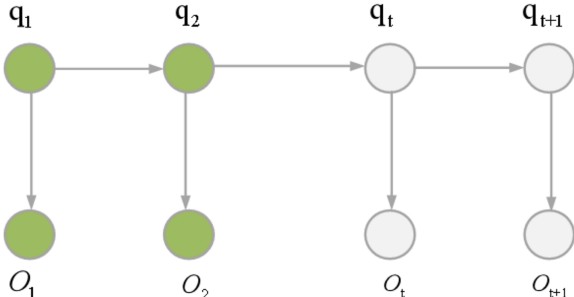

**Figure 4 HMM principle.**

$$q_t \in S = \{s_1, s_2 \ldots s_N\} \tag{2}$$
$$o_t \in V = \{v_1, v_2, \ldots, v_M\} \tag{3}$$

In Eq. (2), S signifies the hidden state set, and N denotes the total number of states. The state of the hidden state chain at time t is represented $q_t \in S$.

A general HMM does not impose any restrictions on the transitions between hidden states, allowing for unrestricted transitions between states. This is referred to as an unconstrained HMM. However, real-world applications often operate within specific industrial contexts. Consequently, an HMM needs to undergo training to align with diverse user requirements. In other words, constraints are typically applied to the HMM or the decoding of hidden state sequences in practical scenarios. Such an HMM is known as a constrained HMM.

This study introduces an HMM that incorporates a constraint related to the minimum continuous length of states, thereby regulating state transitions. Consequently, the constrained HMM is represented as a new five-tuple denoted as $\lambda' = (S', V', A', B', \pi')$. The minimum continuous length of a state, denoted as h, adheres to the constraint $1 \leq h \leq T$.

The hidden state set, denoted as $S'$, is expanded to accommodate $N^*h$ hidden states. In a general HMM, the hidden state set is represented as $s = \{s_1, s_2, \ldots, s_N\}$, and it is expanded by a factor of H. Specifically, $s_1$ is extended to $\{s_{11}, s_{12}, \ldots, s_{1h}\}$; $S_2$ extends to $\{s_{21}, s_{22}, \ldots, s_{2h}\}$; $S_N$ extends to $\{s_{N1}, s_{N2}, \ldots, s_{Nh}\}$. In the output probability matrix and during the analysis of the state chain, $\{s_{11}, s_{12}, \ldots, s_{1h}\}$ is considered equivalent to $s_1$. By extension, the state of the hidden state chain within the expanded set $S' = \{s_{11}, s_{12}, \ldots, s_{1h}, \ldots, s_{Nh}\}$ is represented as $q_t$, where $q_t \in S'$. Figure 5 exhibits the construction of this extended hidden state set.

The observation value set, denoted as $V'$, encompasses M possible observations corresponding to each hidden state, resulting in a total of M observations. $V'$ is equivalent to the standard observation value set $V = \{v_1, v_2, \ldots, v_M\}$. Assuming the observation value at time t is $o_t$, then it follows that $o_t \in V'$. In essence, the observation value set remains consistent with that of a general HMM.

Vector $\pi'$, representing the initial state probabilities, is defined as follows: $\pi = \{\pi_1, \ldots, \pi_{1h}, \ldots, \pi_{Nh}\}$. In this notation, $\pi_{im}$ signifies $P(q_1 = s_{im})$, where $1 \leq i \leq N$. It quantifies the likelihood of the initial state of the sequence being $s_{im}$.

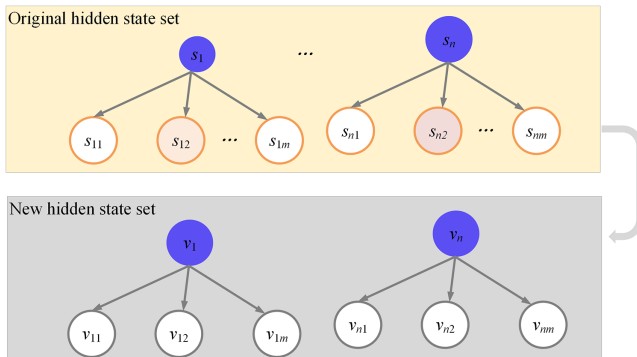

**Figure 5** **Construction of hidden state set.**

In the hidden state transition probability matrix $A'$, denoted as $A' = (a_{im,jn})_{Nh \times Nh}$, the elements $a_{im,jn}$, signify the probability of transition from state $s_{im}$ to $s_{jn}$, expressed as $P(q_{t+1} = s_{jn}|q_t = s_{im})$, where $1 \leq i, j \leq N$, and $1 \leq m, n \leq h$. It is important to note that the constraints imposed on state transitions necessitate that, distinguishing it from the general HMM. To illustrate this distinction, consider the case of having two states and h = 3. In this scenario, the state transition matrix is represented as shown in Eq. (4).

$$A' = \begin{bmatrix} 0 & a_{11,12} & 0 & 0 & 0 & 0 \\ 0 & 0 & a_{12,13} & 0 & 0 & 0 \\ 0 & 0 & a_{13,13} & a_{13,21} & 0 & 0 \\ 0 & 0 & 0 & 0 & a_{21,22} & 0 \\ 0 & 0 & 0 & 0 & 0 & a_{22,23} \\ a_{23,11} & 0 & 0 & 0 & 0 & a_{23,23} \end{bmatrix} \tag{4}$$

Figure 6 outlines the general steps involved in constructing a constrained HMM. Subsequently, building upon the principles of discrete HMM theory, a time series model based on the constrained HMM is developed. The initial step in this process involves discretizing the time series into $o_t$, tailored to the specific requirements of the application context.

In the context of PLTI, this article utilizes the HMM algorithm to tackle the recognition problem. The HMM model, which captures the statistical relationship between hidden states and observable sequences, facilitates the modeling and inference of topological states within power systems. To better align the HMM model with the characteristics of real power systems, a constraint is introduced based on the minimum continuous state length associated with change points. This constraint serves to expand the number of hidden states in the state set and imposes restrictions on the state transition matrix. These adjustments aid in interpreting node connectivity in the power network, ultimately enhancing the accuracy and reliability of topology recognition.

The application of the HMM to data analysis and topological inference in PLTI enables intelligent management and operational support for low-voltage distribution substation areas. The HMM model serves as a valuable tool for gaining a deeper understanding and analysis of the topological characteristics and variations among nodes within the power

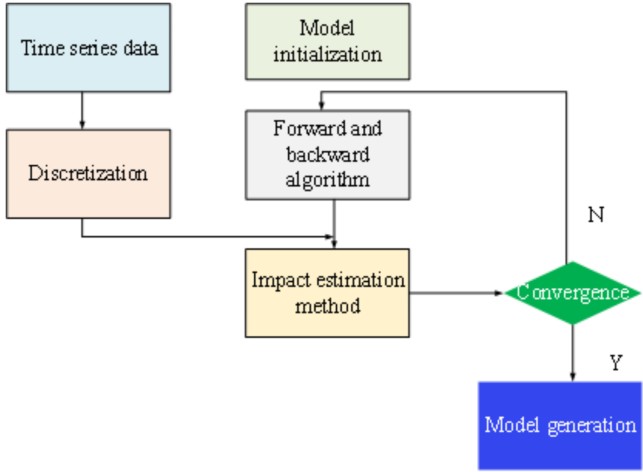

**Figure 6 Model optimization steps.**

network. When integrated with other techniques, the HMM model enhances the accuracy of topology recognition, offering critical support for system operation and management.

This section lays the theoretical foundation and provides methodological support for applying the HMM model in PLTI. The following section presents a specific application scenario, emphasizing the composition and functionality of the LVDN substation identification system. Utilizing the HMM model for data analysis and topology recognition within this system enables intelligent management and operational support for low-voltage distribution substations.

## Composition and function of the LVDN-oriented substation topology identification system

The LVDN substation identification system is a comprehensive system that encompasses multiple electrical components and branches, all meticulously designed to accurately determine the topology of LVDN substations. These components operate collaboratively to ensure the seamless functioning of the distribution substation.

At the core of this system lies the power source, a pivotal component responsible for providing energy. It generates electricity and distributes it to various electrical devices. The power source can take the form of either a conventional generator or a contemporary integrated renewable energy device.

The bus assumes a crucial role within the system as the primary conductor, interconnecting various electrical devices. Its principal function is to transmit electrical energy from the power source to the substation's components while regulating the voltage level to meet the requirements of different devices. The bus is connected to components such as generators, transformers, and switches through branches, forming the primary pathway for electrical energy transmission.

Transformers hold a central position in the system, primarily responsible for voltage level adjustment. They facilitate the conversion of high voltage to low voltage or vice versa, ensuring compatibility with the operational needs of diverse devices. Interconnected with

the bus and feeder lines *via* branches, transformers efficiently regulate the transmission and distribution of electrical energy.

Switches are fundamental devices essential for controlling the flow of electric current. Their capacity to either open or close circuits is pivotal in the distribution and regulation of electricity. Within the context of the LVDN substation identification system, switches assume a central role in directing electrical energy to various branches and feeder lines. This functionality is critical for effectively managing and controlling the flow of electricity throughout the system.

Feeder lines play a crucial role in the transmission of electrical energy from the transformer to end-users within the substation. Their primary responsibility is to distribute electricity to a variety of electrical devices and users, guaranteeing a continuous and uninterrupted power supply. These feeder lines achieve this by establishing connections between transformers and switches through branches, thus forming the vital conduit for the transmission of electrical energy.

Branches serve as vital pathways, establishing interconnections among various electrical components. They facilitate the flow of electrical energy, originating from the power source, through the bus, transformers, switches, and other elements, before finally reaching the end-users. The design and operation of branches play a pivotal role in determining the efficiency and reliability of the substation identification system. Well-designed branches are instrumental in ensuring the effective transmission and stable provision of electrical energy.

# CASE ANALYSIS

## Experimental design

(1) Data source

This section conducts a case analysis using user data obtained from a substation under the jurisdiction of X city, collected by the state grid Power consumption information acquisition system. The input data consists of three days' worth of single-phase user voltage measurements, obtained from the historical records of the substation, with a data collection rate of 100%. The substation serves 125 single-phase users, distributed across 16 secondary access meter boxes operating under three-phase configurations, with no three-phase users present. Each user's dataset comprises 190 voltage measurements. The initial phase of the experiment focuses on identifying the user phase and access meter box within the designated low-voltage platform area. Subsequently, the method proposed in this article's analysis is compared to other methods in terms of user phase identification accuracy.

(2) Selection of experimental content and evaluation index

In the experiment, the investigation begins with user phase and meter box identification, followed by a comparative analysis of the method against existing approaches to assess user phase identification errors. Subsequently, in line with the specific requirements of power system topology identification tasks and the experimental design, the research examines the influence of adjusting the discretization granularity on user phase identification accuracy. The discretization granularity is varied within the range of

5% to 18%. Next, the algorithm's resistance to noise is evaluated. Various noise levels, characterized by standard deviations of 0.01 and 0.1, are introduced to simulate real-world power data noise. A performance comparison is conducted between the proposed improved HMM algorithm and previously existing algorithms, including the classical HMM algorithm. Finally, the experiment records the execution times of different algorithms to gauge their efficiency. Additionally, the assessment covers energy utilization efficiency and load balancing to understand their impact on the power system.

### Error analysis

Figure 7 illustrates the assessment of user phase identification errors using the HMM algorithm under different discretization granularities. In Fig. 7, the optimized HMM algorithm demonstrates strong convergence on the test data. As the number of iterations increases from 10 to 100, the model error consistently decreases. Notably, the learning speed of the model is influenced by the length of the sample data (time series length). When the time series length is short, the reduction in model error occurs at a slower pace, leading to a more substantial gap between the learned weight matrix and the actual weight matrix. Conversely, with a longer time series length, the model error decreases more rapidly, reaching an optimal solution in as few as 50 iterations. Upon comparing the algorithm introduced in this article with the recent study conducted by *Li et al. (2023)*, it is evident that both algorithms exhibit a comparable number of iterations. This similarity in iteration counts indicates that the algorithm presented in this article efficiently attains an optimal solution, aligning with the observations made in the study by *Li et al. (2023)*.

### Verification and comparison of algorithm anti-noise performance

Figure 8 presents a comparative evaluation between the standard HMM algorithm and the proposed optimized HMM algorithm, denoted as "HMM (1)," at noise standard deviations of 0.01 and 0.1. HMM (1) in Fig. 8 is the optimized HMM algorithm.

Figure 8 illustrates the considerable superiority of the enhanced HMM algorithm, especially when dealing with a substantial volume of sample data. In contrast to the conventional HMM algorithm, the improved variant excels in scenarios with a higher number of samples, showcasing enhanced learning capabilities and reduced errors. These findings suggest that the enhanced HMM algorithm is well-suited for achieving more precise topology identification of power lines in real-world applications.

### Algorithm comparison

Table 1 demonstrates that the enhanced HMM method, when compared to full-day data analysis, introduces only a modest increase in computational time while significantly elevating the accuracy of the discrimination process. Although the sliding time window analysis method exhibits the highest precision, its computational time is 50 times that of the improved HMM method, which could become prohibitively time-consuming in the context of a large-scale power grid. Nevertheless, the enhanced HMM method exhibits substantial potential for practical applications within this domain. To address the issue of threshold selection while maintaining high precision, a dual-threshold verification strategy

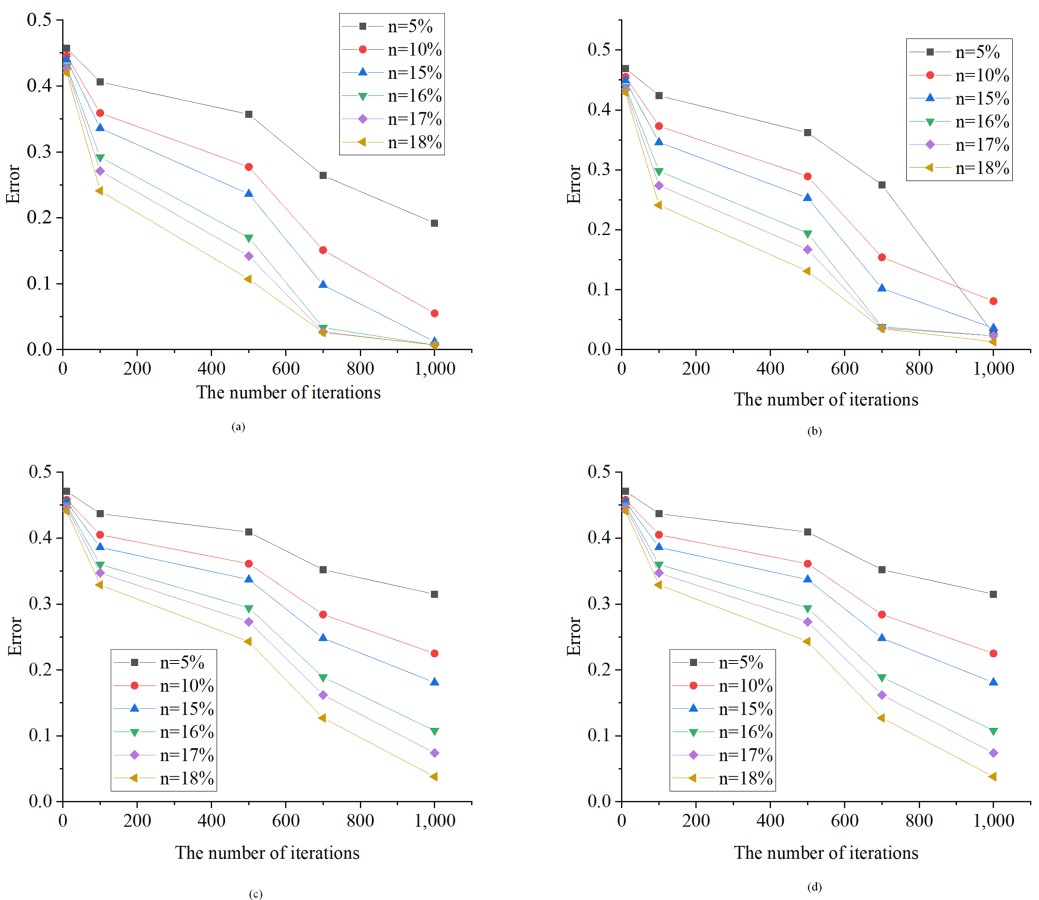

**Figure 7 Error test results of the HMM algorithm.** *n*, Discretization granularity. (A) 50 samples; (B) 100 samples; (C) 150 samples; (D) 190 samples.

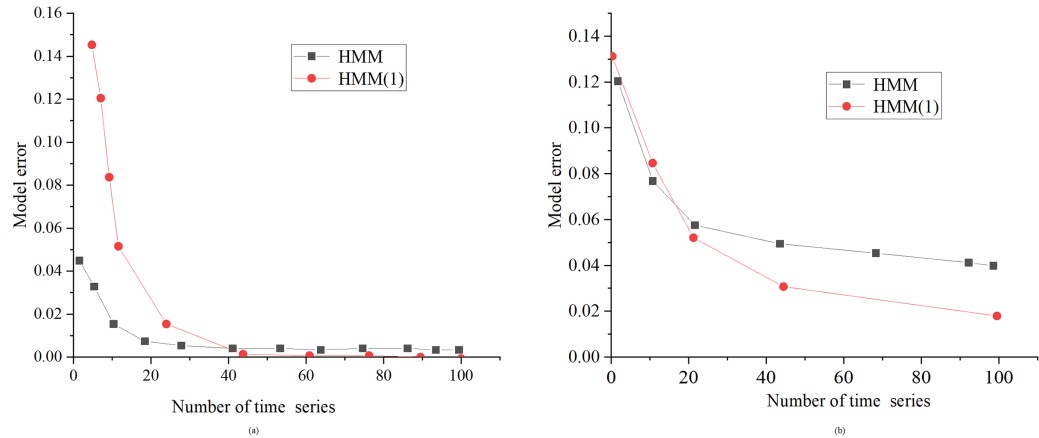

**Figure 8 Performance comparison between HMM algorithm and the proposed optimized HMM algorithm under the same noise.** (A) SD of noise is 0.01; (B) SD of noise is 0.1.

**Table 1 Comparison of the running time of different algorithms.**

| Algorithms | Time consumption/ms | Accuracy (0.9) | Accuracy (0.95) |
|---|---|---|---|
| Full day data analysis | 16.1 | No | No |
| Sliding time window analysis | 1,743.2 | Yes | Yes |
| The proposed improved HMM algorithm | 35.1 | Yes | No |

**Table 2 Comparison of algorithm energy efficiency and load balancing index of the algorithms.**

| Algorithms | Energy utilization efficiency | Load balancing |
|---|---|---|
| Full day data analysis | 0.75 | 0.62 |
| Sliding time window analysis | 0.82 | 0.78 |
| The proposed improved HMM algorithm | 0.89 | 0.85 |

can be employed, particularly well-suited for user data with precision levels ranging from 0.90 to 0.95. This approach is designed to mitigate the risk of misclassifications while efficiently managing computational resources.

The algorithm performance evaluation in this article includes a comparative analysis of energy efficiency and load balance. Table 2 presents the energy utilization efficiency and load balancing degree of the three algorithms.

Table 2 provides insights into different algorithms' energy utilization efficiency and load balancing performance. The results show that the full-day data analysis algorithm exhibits an energy utilization efficiency of 0.75, slightly outperformed by the sliding time window analysis algorithm, which achieves an efficiency of 0.82. In contrast, the enhanced HMM algorithm, developed based on the power grid topology model presented in this study, demonstrates superior energy utilization efficiency, with a value of 0.89. These findings indicate that the improved HMM algorithm is more adept at optimizing energy resource utilization within the power system. In terms of load balancing, the full-day data analysis algorithm yields a load balance index of 0.62, while the sliding time window analysis algorithm achieves a higher index of 0.78. Notably, the enhanced HMM algorithm attains the highest load balance index at 0.85. This underscores the enhanced HMM algorithm's capacity to excel in load balancing within the power system, ensuring a more equitable distribution of loads among different nodes. As a result, the improved HMM algorithm contributes to enhanced stability and reliability in the power system.

## DISCUSSION

The enhanced HMM algorithm presents distinct advantages in terms of user phase recognition accuracy, noise resilience, and computational efficiency. As the granularity of discretization increases, there is a gradual reduction in model error. This phenomenon is attributed to the finer granularity, which provides a more extensive set of data points, enabling the model to capture user phase change patterns with greater precision. Furthermore, the augmented number of samples bolsters the model's learning capacity,

thereby further elevating recognition accuracy. When assessing the performance of both the HMM algorithm and the enhanced HMM algorithm under varying noise levels, it becomes evident that as noise levels escalate, model errors also increase. However, the enhanced HMM algorithm exhibits superior noise resistance. This advantage arises from the incorporation of more effective noise filtering methods, which curtail noise interference and bolster the model's robustness. In contrast, the conventional HMM algorithm proves to be more susceptible to noise, rendering it susceptible to substantial errors in the presence of interference. In summation, the proposed enhanced HMM algorithm achieves expedited computational times and upholds accuracy, rendering it a more practical and efficient choice for real-world applications.

Based on the experimental findings and discussions presented earlier, the enhanced HMM algorithm demonstrates notable advantages in the domain of PLTI. It excels particularly in terms of time efficiency, accuracy, energy utilization efficiency, and load balancing. Nevertheless, it is crucial to acknowledge potential limitations and potential biases that warrant consideration. Firstly, it is crucial to recognize that the choice of sample size and discretization granularity may vary across different power systems and datasets (*Marin-Quintero, Orozco-Henao & Mora-Florez, 2023*), necessitating further research to validate the algorithm's applicability in diverse contexts. Secondly, while noise resistance has been addressed to some extent, validating its performance under various noise types and intensities is imperative to ensure robustness. Additionally, there is room for further optimization of computational resource requirements, particularly in the context of large-scale power systems. The representativeness of experimental data and the comprehensiveness of algorithm comparison and evaluation also merit careful attention. In conclusion, despite the promising potential of the enhanced HMM algorithm in PLTI, it is essential to underscore that further research and validation in practical applications are imperative to establish the reliability and broad applicability of its performance.

## CONCLUSION

The enhanced HMM algorithm presents significant advantages in the realms of user phase recognition, noise resilience, and computational efficiency. Experimental tests involving various levels of discretization granularity reveal a notable trend: as granularity increases, there is a corresponding reduction in model error. This phenomenon is attributed to finer granularity providing an increased number of data points, thereby facilitating the model in capturing user phase changes with heightened precision. Concurrently, augmenting the number of samples bolsters the model's learning capacity, enhancing recognition accuracy. With respect to noise resistance, the improved HMM algorithm notably outperforms its traditional counterpart. This superiority can be attributed to the incorporation of more effective noise filtering methods, which mitigate noise interference and bolster the model's overall robustness. In contrast, the conventional HMM algorithm proves to be more susceptible to the detrimental effects of noise interference, leading to more substantial errors. In evaluating both performance and computational efficiency, the enhanced HMM algorithm strikes a balance between accuracy and computational swiftness. Through strategic algorithm design and optimization of the calculation process, it attains shorter

computation times, thereby augmenting its practical feasibility and efficiency. Future research endeavors will delve deeper into the application of this improved algorithm within domains related to the principles of energy conservation and the assessment of sensor micro-energy harvesting technologies.

### Funding
The authors received no funding for this work.

### Competing Interests
Jianfeng Ji is employed by Beijing Zhixiang Technology Co., Ltd and Yongjin Xu, Jifan Lv, Jiaying Wang, Fangbin Ye and Shen Ye are employed by the State grid Zhejiang marketing service center. The authors declare that they have no competing interests.

### Author Contributions
- Yongjin Xu conceived and designed the experiments, analyzed the data, authored or reviewed drafts of the article, and approved the final draft.
- Jifan Lv conceived and designed the experiments, analyzed the data, prepared figures and/or tables, and approved the final draft.
- Jiaying Wang performed the experiments, analyzed the data, prepared figures and/or tables, and approved the final draft.
- Fangbin Ye performed the experiments, performed the computation work, authored or reviewed drafts of the article, and approved the final draft.
- Shen Ye conceived and designed the experiments, performed the computation work, prepared figures and/or tables, and approved the final draft.
- Jianfeng Ji performed the experiments, performed the computation work, authored or reviewed drafts of the article, and approved the final draft.

### Data Availability
The raw data are available in the Supplemental Files.

### Supplemental Information
Supplemental information for this article can be found online at http://dx.doi.org/10.7717/peerj-cs.1688#supplemental-information.

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
