# Peer review of "Identifying topology of distribution substation in power Internet of Things using dynamic voltage load fluctuation flow analysis"

_PeerJ Computer Science, doi:10.7717/peerj-cs.1688_

## Round 0.1 · original submission · Major Revisions

Please revise your manuscript according to the comments of reviewers.

Reviewer 1 ·

Basic reporting

Errors in references citation. the content of literature 2 has no connection with the theme of the article...
The pictures in the article are blurry and unclear, the optimization goals are not clear, and the necessary academic issues are not reflected in the paper.

Experimental design

Insufficient

Validity of the findings

no comment

Additional comments

no comment

Reviewer 2 ·

Basic reporting

The advantage of this paper aims to adapt the TSS algorithm better to practical applications by proposing a TSS algorithm based on the constrained Hidden Markov Model (HMM) The proposed algorithm simplifies change point detection, enhances the algorithm's scalability, and presents better adaptability This work analyzes the typical topology of the Low Voltage Distribution Network (LVDN) Substation and implements the LVDN Substation oriented topology identity based on the time series of electric consumption measurement and the law of electric energy conservation The sliding time window analysis method has the
Highest accuracy How, its time consumption is 50 times that of the proposed method.
Ensure that the abstract clearly and accurately summarizes the purpose, methods, main results, and conclusions of the paper. For example, in the abstract, it is mentioned that “Thus, the proposed model can converge quickly to obtain the optimal solution. Meanwhile, the proposed method does not increase calculation time much while effectively improving the accuracy of the discrimination process”. The statement here is very general and should be concretized.
The first part of the method involves many basic concepts and theoretical knowledge. Are these concepts necessary for the research topic? It can be merged into the background section of the introduction, which should be practical solutions related to the topic, rather than some concepts.
The figures and tables in the paper should be clear and orderly, but Figure 1 is very blurry, which greatly affects the overall quality of the paper. There was no further textual explanation of the elements and composition in the figure, which made the understanding very difficult.
What is the meaning of 5% -18% in the legend section of Figure 8- Figure 10? Why not distribute it in equal proportions? And there is no mention of these ratios in the text, which seems very confusing.

Experimental design

Section 2.3 establishes a Hidden Markov model. What is the relationship between the model and the subject? What is the connection between it and the context? At present, it looks like an independent part. Hence, it's necessary to have a strong connection between which part to go and the theme.

Validity of the findings

The results section is not rich enough, especially the explanation of image parameters and values is very simple, and there is a lack of further in-depth discussion. It is necessary to compare with recent similar studies to highlight the advantages and reliability of this study.
What is the difference between HMM and HMM (1) in Figure 11? Is there a specific explanation in the text? Is it a new model or the model mentioned earlier?

Additional comments

In addition, the review of existing literature in the introduction only stays in 2021 and does not cover the research results from 2022 and 2023. You can add three more articles to review the latest research status. Without this, simply listing the content of these literature reviews to summarize the current industry situation is far from enough. What is important is what information can be extracted from these achievements? For example, what are the limitations and research gaps that this article can fill? These contents need to be summarized at the end.The conclusion section did not accurately summarize the main limitations and future research directions of this study. In addition, the description of the main research contributions is not clear enough, and it is best to list the innovative achievements of this study one by one.

Reviewer 3 ·

Basic reporting

Most traditional Substation Topology identification systems use power line carrier communication, which uses spread spectrum, frequency modulation, intermediate frequency, and audio carrier to transmit signals. In these circumstances, the capacitance and inductance in the power grid significantly impact the carrier signal, resulting in unstable transmission. This work proposes a voltage correlation analysis method based on dynamic Time Series Segmentation (TS) for LVDN Substation Topology identification. The proposed method uses the Extreme Point Determination (EPD) method to impact the Time Series (TS) change. On this basis, the principle of voltage correlation is applied to judge the voltage correlation of different sequences, including extreme points, and identify the Substation Topology.

Clear, unambiguous, professional English language used throughout.
-Yes.
Intro & background to show context. Literature well referenced & relevant.
-Yes.
Structure conforms to PeerJ standards, discipline norm, or improved for clarity.
-Yes.
Figures are relevant, high quality, well labelled & described.
- Please improve the clarity and quality of the image.

Experimental design

Original primary research within Scope of the journal.
-Yes.
Research question well defined, relevant & meaningful. It is stated how the research fills an identified knowledge gap.
-Yes.
Rigorous investigation performed to a high technical & ethical standard.
-No.
Methods described with sufficient detail & information to replicate.
-No. Can be more detailed.

Validity of the findings

Impact and novelty not assessed. Meaningful replication encouraged where rationale & benefit to literature is clearly stated.
-Please specify the novelty and contribution of the manuscript.
All underlying data have been provided; they are robust, statistically sound, & controlled.
-No. Please provide raw data.
Conclusions are well stated, linked to original research question & limited to supporting results.
-Yes.

Additional comments

A. "this paper aims to adapt the TSS algorithm" appears in line 90. What is the relationship between the TSS algorithm and the TS algorithm mentioned in lines 85-86? Is it obtained through optimization or other irrelevant algorithms?
B. Update the research of scholars in the field of power network or voltage topology identification in the introduction to show the latest relevant research. At the same time, the advantages and disadvantages of the literature should be described to highlight the innovation of this research.
C. It is suggested to give a specific description of the PIoT architecture in Figure 2. For example, how does NCT provide support for PIoT? How sensor monitoring information is processed?
D. You should elaborate on the operation mode of each component and branch of the substation topology identification system in Section 2.4 in more detail, reflecting the advantages of the power system.
E. Please optimize the pictures in Figure 1-Figure 12 in the manuscript to further improve the resolution.
F. In the substation topology identification system mentioned in Section 2.3, how is the HMM algorithm applied? The author is required to add a description.
G. The number of samples in Figure 7 mainly includes 50/100/150/200, but is there any correlation between the number of users and the number of voltages appearing in lines 241-242? Please explain.
H. When analyzing the performance of the power grid topology model designed for this research, the results shown in the manuscript are too thin. You can add other indicators, such as energy efficiency, load balance, etc.
I. The conclusion section should give a detailed description of the results and contributions of this study on the substation system. The author is suggested to add this information.

---

## Round 0.2 · Minor Revisions

1. Please further clarify the contributions in this article.
2. Another reason the paper needs to be revised is the language used in the paper. There are not many obvious grammatical errors in the paper, but one can find many sentences whose meanings are not clear.

**Language Note:** The Academic Editor has identified that the English language must be improved. PeerJ can provide language editing services - please contact us at [email protected] for pricing (be sure to provide your manuscript number and title). Alternatively, you should make your own arrangements to improve the language quality and provide details in your response letter. – PeerJ Staff

Reviewer 2 ·

Basic reporting

This paper proposes an improved method for identifying the topology of distribution substations using Hidden Markov Models. The approach mitigates the impact of power line restructuring, maintenance, and inspections on power grid stability and energy metering accuracy. Experimental results validate the effectiveness of the proposed model, which offers valuable insights for ensuring stability in power grid operations and enhancing energy metering accuracy. While the current study provides more substantial findings, there are some issues with the article that need to be revised.
1.The innovative aspects of the paper could be highlighted and it could be made clear how it is differentiated from existing studies so that the reader can better understand the unique contribution of the study.
2.It is proposed to provide more background information for the introduction section, describing the importance of grid topology identification and the current challenges. The significance and potential impact of this research on achieving stable operation of smart distribution networks are described.
3.In the section synthesizing the related studies, some discussion about the shortcomings of the existing methods can be added and the innovations of your own research can be presented. You should emphasize how the proposed methodology addresses the limitations of current methods and provides higher accuracy and efficiency.

Experimental design

1.In Section 3.1, the explanation of discretization granularity can be simplified to directly state that the choice of discretization particle size is based on specific requirements and experimental design and provide the range of discretization particle sizes chosen. There is no need to elaborate on the concepts and principles of discretization granularity.
2. In Section 3.2, the description of the validation and comparison of the noise performance can be simplified. It is straightforward to state that the optimized HMM algorithm performs better in noisy environments and provide the necessary graphical support. There is no need to explain too much about the differences between the HMM algorithm and the optimized HMM algorithm.

Validity of the findings

1.In Section 3, it is recommended that the algorithms used and the experimental results be described in a clear and concise manner. A succinct summary can be provided to show that the proposed improved algorithm has higher accuracy and efficiency compared to other methods. Also, when presenting the experimental results, only key data and graphs should be provided to avoid lengthy descriptions and repetitive content.
2. In the conclusion section, the description of the contributions and limitations of the improved method can be simplified. It is straightforward to suggest that the improved HMM algorithm has outstanding performance in terms of user phase identification, anti-noise performance and computational efficiency, and to point out that future research can further explore the applications in terms of linear energy conservation laws and sensor micro-energy harvesting and evaluation techniques. A detailed explanation of the theoretical framework and details of the algorithm is not required.

Additional comments

1.In Section 2.1, it is recommended to simplify the introduction to IoT and Power IoT. It could start with an overview of the definition and application areas of IoT and then go directly to Power IoT and its role in the power system. In addition, specific examples or case studies could be provided to illustrate the significance and benefits of Power IoT.
2. In Section 2.2, it is recommended to simplify the description of time series analysis and voltage correlation. The fundamentals of time series analysis and voltage correlation can be presented by providing concise definitions and equations. Emphasis is placed on voltage correlation as a quantitative metric for determining topology, and it is noted that commonly used metrics are Euclidean distance, dynamic time regularization, and Pearson correlation coefficient.
3.In the discussion section, the experimental results can be succinctly summarized. It is pointed out that the improved HMM algorithm has significant advantages in terms of user phase recognition accuracy, anti-noise performance, and computational efficiency. It is not necessary to discuss the details of each experimental result with too much description.

Reviewer 3 ·

Basic reporting

a) At the beginning of the abstract and introduction, you should clearly introduce why the reorganization, maintenance, and inspection of power lines are crucial to the stability of power grid operation and the accuracy of energy metering. This helps to provide readers with clearer background information.
b) Adding a literature review section may help readers better understand the development of the research field of the article, thus enhancing the persuasiveness of the article.

Experimental design

c) You mentioned that the experimental results verified the validity of the model, but in fact, you didn't provide more specific details of the experiment. For example, the detailed design of the experimental process, the source of the data set, and the selection of evaluation indicators. The supplement of this information will help readers to evaluate the repeatability and credibility of the experiment.
d) The content of the experimental design section is too little, please expand it, and it is best to describe it in several subsections.

Validity of the findings

e) In order to verify the efficiency of the proposed model, in fact, it is necessary to compare this model with all other models of the same type.
f) For your model, you should also discuss the potential limitations and possible deviations. For example, have you considered all possible situations in the experiment, or are there possible systematic errors?

Additional comments

g) Please expand the number of references to keep it at about 35, so as to reflect the scientific nature of the article.
h) Figure 5 shows too little information, please adjust it, including adding text.

---

## Round 0.3 · accepted · Accept

I think that you have addressed all of the reviewers' comments, and this manuscript is ready for publication.